# Algorithms and Theory for Supervised Gradual Domain Adaptation

**Jing Dong**                                              *jingdong@link.cuhk.edu.cn*
*The Chinese University of Hong Kong, Shenzhen*

**Shiji Zhou**                                             *zhoushiji00@gmail.com*
*Tsinghua University*

**Baoxiang Wang**                                          *bxiangwang@cuhk.edu.cn*
*The Chinese University of Hong Kong, Shenzhen*

**Han Zhao**                                               *hanzhao@illinois.edu*
*University of Illinois Urbana-Champaign*

**Reviewed on OpenReview:** *https://openreview.net/forum?id=35y5hv9fbb*

## Abstract

The phenomenon of data distribution evolving over time has been observed in a range of applications, calling for the need for adaptive learning algorithms. We thus study the problem of supervised gradual domain adaptation, where labeled data from shifting distributions are available to the learner along the trajectory, and we aim to learn a classifier on a target data distribution of interest. Under this setting, we provide the first generalization upper bound on the learning error under mild assumptions. Our results are algorithm agnostic, general for a range of loss functions, and only depend linearly on the averaged learning error across the trajectory. This shows significant improvement compared to the previous upper bound for unsupervised gradual domain adaptation, where the learning error on the target domain depends exponentially on the initial error on the source domain. Compared with the offline setting of learning from multiple domains, our results also suggest the potential benefits of the temporal structure among different domains in adapting to the target one. Empirically, our theoretical results imply that learning proper representations across the domains will effectively mitigate learning errors. Motivated by these theoretical insights, we propose a min-max learning objective to learn the representation and classifier simultaneously. Experimental results on both semi-synthetic and large-scale real datasets corroborate our findings and demonstrate the effectiveness of our objectives.

## 1 Introduction

An essential assumption for the deployment of machine learning models in real-world applications is the alignment of training and testing data distributions. Under this condition, models are expected to generalize, yet real-world applications often fail to meet this assumption. Instead, continual distribution shift is widely observed in a range of applications. For example, satellite images of buildings and lands change over time due to city development (Christie et al., 2018); self-driving cars receive data with quality degrading towards nightfall (Bobu et al., 2018; Wu et al., 2019b). Although this problem can be mitigated by collecting training data that covers a wide range of distributions, it is often impossible to obtain such a large volume of labeled data in many scenarios. On the other hand, the negligence of shifts between domains also leads to suboptimal performance. Motivated by this commonly observed phenomenon of gradually shifting distributions, we study supervised gradual domain adaptation in this work. Supervised gradual domain adaptation models the training data as a sequence of batched data with underlying changing distributions, where the ultimate

goal of learning is to obtain an effective classifier on the target domain at the last step. This relaxation of data alignment assumption thus equips gradual domain adaptation with applicability in a wide range of scenarios. Compared with unsupervised gradual domain adaptation, where only unlabeled data is available along the sequence, in supervised gradual domain adaptation, the learner also has access to labeled data from the intermediate domains. Note that this distinction in terms of problem setting is essential, as it allows for more flexible model adaptation and algorithm designs in supervised gradual domain adaptation.

The mismatch between training and testing data distributions has long been observed, and it had been addressed with conventional domain adaptation and multiple source domain adaptation (Duan et al., 2012; Hoffman et al., 2013; 2018b;a; Zhao et al., 2018; Wen et al., 2020; Mansour et al., 2021) in the literature. Compared with the existing paradigms, supervised gradual domain adaptation poses new challenges for these methods, as it involves more than one training domains and the training domains come in sequence. For example, in the existing setting of multiple-source domain adaptation (Zhao et al., 2018; Hoffman et al., 2018a), the learning algorithms try to adapt to the target domain in a one-off fashion. Supervised gradual domain adaptation, however, is more realistic, and allows the learner to take advantage of the temporal structure among the gradually changing training domains, which can lead to potentially better generalization due to the smaller distributional shift between each consecutive pair of domains.

Various empirically successful algorithms have been proposed for gradual domain adaptation (Hoffman et al.; Gadermayr et al., 2018; Wulfmeier et al., 2018; Bobu et al., 2018). Nevertheless, we still lack a theoretical understanding of their limits and strengths. The first algorithm-specific theoretical guarantee for unsupervised gradual domain adaptation is provided by Kumar et al. (2020). However, the given upper bound of the learning error on the target domain suffers from exponential dependency (in terms of the length of the trajectory) on the initial learning error on the source domain. This is often hard to take in reality and it is left open whether this can be alleviated in supervised gradual domain adaptation.

In this paper, we study the problem of gradual domain adaptation under a supervised setting where labels of training domains are available. We prove that the learning error of the target domain is only linearly dependent on the averaged error over training domains, showing a significant improvement compared to the unsupervised case. We show that our results are comparable with the learning bound for multiple source training and can be better under certain cases while relaxing the requirement of access to all training domains upfront simultaneously. Further, our analysis is algorithm and loss function independent. Compared to previous theoretical results on domain adaptation, which used $l_1$ distance (Mansour et al., 2009) or $W_\infty$ distance to capture shifts between data distributions (Kumar et al., 2020), our results are obtained under milder assumptions. We use $W_p$ Wasserstein distance to describe the gradual shifts between domains, enabling our results to hold under a wider range of real applications. Our bound features two important ingredients to depict the problem structure: sequential Rademacher complexity (Rakhlin et al., 2015) is used to characterize the sequential structure of gradual domain adaptation while discrepancy measure (Kuznetsov & Mohri, 2017) is used to measure the non-stationarity of the sequence.

Our theoretical results provide insights into empirical methods for gradual domain adaptation. Specifically, our bound highlights the following two observations: (1) Effective representation where the data drift is "small" helps. Our theoretical results highlight an explicit term showing that representation learning can directly optimize the learning bound. (2) There exists an optimal time horizon (number of training domains) for supervised gradual domain adaptation. Our results highlight a trade-off between the time horizon and the learning bound.

Based on the first observation, we propose a min-max learning objective to learn representations concurrently with the classifier. Optimizing this objective, however, requires simultaneous access to all training domains. In light of this challenge, we relax the requirement of simultaneous access with temporal models that encode knowledge of past training domains. To verify our observations and the proposed objectives, we conduct experiments on both semi-synthetic datasets with MNIST dataset and large-scale real datasets such as FMOW (Christie et al., 2018). Comprehensive experimental results validate our theoretical findings and confirm the effectiveness of our proposed objective.

## 2 Related Work

**(Multiple source) domain adaptation** Learning with shifting distributions appears in many learning problems. Formally referred as domain adaptation, this has been extensively studied in a variety of scenarios, including computer vision (Hoffman et al.; Venkateswara et al., 2017; Zhao et al., 2019b), natural language processing (Blitzer et al., 2006; 2007; Axelrod et al., 2011), and speech recognition (Sun et al., 2017; Sim et al., 2018). When the data labels of the target domain are available during training, known as supervised domain adaptation, several parameter regularization-based methods (Yang et al., 2007; Aytar & Zisserman, 2011), feature transformations based methods (Saenko et al., 2010; Kulis et al., 2011) and a combination of the two are proposed (Duan et al., 2012; Hoffman et al., 2013). On the theoretical side of domain adaptation, Ben-David et al. (2010) investigated the necessary assumptions for domain adaptations, which specifies that either the domains are needed to be similar, or there exists a classifier in the hypothesis class that can attain low error on both domains. For the restriction on the similarity of domains, Wu et al. (2019a) proposed an asymmetrically-relaxed distribution alignment for an alternative requirement. This is a more general condition for domain adaptation when compared to those proposed by Ben-David et al. (2010), and holds in a more general setting with high-capacity hypothesis classes such as neural networks. Zhao et al. (2019a) then focused on the efficiency of representation learning for domain adaption and characterized a trade-off between learning an invariant representation across domains and achieving small errors on all domains. This is then extended to a more general setting by Zhao et al. (2020), which provides a geometric characterization of feasible regions in the information plane for invariant representation learning. The problem of adapting with multiple training domains, referred to as multiple source domain adaptation (MDA), is also studied extensively. The first asymptotic learning bound for MDA is studied by Hoffman et al. (2018a). Follow-up work Zhao et al. (2018) provides the first generalization bounds and proposed efficient adversarial neural networks to demonstrate empirical superiority. The theoretical results are further explored by Wen et al. (2020) with a generalized notion of distance measure, and by Mansour et al. (2021) when only limited target labeled data are available.

**Gradual domain adaptation** Many real-world applications involve data that come in sequence and are continuously shifting. This first attempt addresses the setting where data from continuously evolving distribution with a novel unsupervised manifold-based adaptation method (Hoffman et al.). Following works Gadermayr et al. (2018); Wulfmeier et al. (2018); Bobu et al. (2018) also proposed unsupervised approaches for this variant of gradual domain adaptation with unsupervised algorithms. The first to study the problem of adapting to an unseen target domain with shifting training domains is Kumar et al. (2020). Their result features the first theoretical guarantee for unsupervised gradual domain adaptation with a self-training algorithm and highlights that learning with a gradually shifting domain can be potentially much more beneficial than a Direct Adaptation. The work provides a theoretical understanding of the effectiveness of empirical tricks such as regularization and label sharpening. However, they are obtained under rather stringent assumptions. They assumed that the label distribution remains unchanged while the varying class conditional probability between any two consecutive domains has bounded $W_\infty$ Wasserstein distance, which only covers a limited number of cases. Moreover, the loss functions are restricted to be the hinge loss and ramp loss while the classifier is restricted to be linear. Under these assumptions, the final learning error is bounded by $O(\exp(T)\alpha_0)$, where $T$ is the horizon length and $\alpha_0$ is the initial learning error on the initial domain. This result is later extended by Chen et al. (2020) with linear classifiers and Gaussian spurious features and improved by concurrent independent work Wang et al. (2022) to $O(\alpha_0 + T)$ in the setting of unsupervised gradual domain adaptation. The theoretical advances are complemented by recent empirical success in gradual domain adaptation. Recent work Chen & Chao (2021) extends the unsupervised gradual domain adaptation problem to the case where intermediate domains are not already available. Abnar et al. (2021) and Sagawa et al. (2021) provide the first comprehensive benchmark and datasets for both supervised and unsupervised gradual domain adaptation.

## 3 Preliminaries

The problem of gradual domain adaptation proceeds sequentially through a finite time horizon $\{1, \ldots, T\}$ with evolving data domains. A data distribution $P_t \in \mathbb{R}^d \times \mathbb{R}^k$ is realized at each time step with the features

denoted as $X \in \mathbb{R}^d$ and labels as $Y \in \mathbb{R}^k$. With a given loss function $\ell(\cdot, \cdot)$, we are interested in obtaining an effective classifier $h \in \mathcal{H} : \mathbb{R}^d \to \mathbb{R}^k$ that minimizes a given loss function on the target domain $P_T$, which is also the last domain. With access to only $n$ samples from each intermediate domain $P_1, \ldots, P_{T-1}$, we seek to design algorithms that output a classifier at each time step where the final classifier performs well on the target domain.

Following the prior work (Kumar et al., 2020), we assume the shift is gradual. To capture such a gradual shift, we use the Wasserstein distance to measure the change between any two consecutive domains. The Wasserstein distance offers a way to include a large range of cases, including the case where the two measures of the data domains are not on the same probability space (Cai & Lim, 2020).

**Definition 3.1.** *(Wasserstein distance) The p-th Wasserstein distance, denoted as $W_p$ distance, between two probability distribution $P, Q$ is defined as $W_p(P, Q) = \left( \inf_{\gamma \in \Gamma(P,Q)} \int \|x - y\|^p d\gamma(x, y) \right)^{1/p}$ , where $\Gamma(P, Q)$ denotes the set of all joint distribution $\gamma$ over $(X, Y)$ such that $X \sim P$, $Y \sim Q$.*

Intuitively, Wasserstein distance measures the minimum cost needed to move one distribution to another. The flexibility of Wasserstein distance enables us to derive tight theoretical results for a wider range of practical applications. In comparison, previous results leverage $l_1$ distance (Mansour et al., 2009) or the Wasserstein-infinity $W_\infty$ distance (Kumar et al., 2020) to capture non-stationarity. In practice, however, this is rarely used and $W_1$ is more commonly employed due to its low computational cost. Moreover, even if pairs of data distributions are close to each other in terms of $W_1$ distance, the $W_\infty$ distance can be unbounded with a few presences of outlier data. Previous literature hence offers limited insights whereas our results include this more general scenario. We formally describe the assumptions below.

**Assumption 3.1.** *For all $1 \leq t \leq T - 1$ and some constant $\Delta > 0$, the p-th Wasserstein distance between two domains is bounded as $W_p(P_t(X, Y), P_{t+1}(X, Y)) \leq \Delta$.*

We study the problem without restrictions on the specific form of the loss function, and we only assume that the empirical loss function is bounded and Lipschitz continuous. This covers a rich class of loss functions, including the logistic loss/binary cross-entropy, and hinge loss. Formally, let $\ell_h$ be the loss function, $\ell_h = \ell(h(x), y) : \mathcal{X} \times \mathcal{Y} \to \mathbb{R}$. We have the following assumption.

**Assumption 3.2.** *The loss function $\ell_h : \mathcal{X} \times \mathcal{Y} \to \mathbb{R}$ is $\rho$-Lipschitz continuous with respect to $(x, y)$, denoted as $\|\ell_h\|_{Lip} \leq \rho$, and bounded such that $\|\ell_h\|_\infty \leq M$.*

This assumption is general as it holds when the input data are compact. Moreover, we note that this assumption is mainly for the convenience of technical analysis and is common in the literature (Mansour et al., 2009; Cortes & Mohri, 2011; Kumar et al., 2020).

The gradual domain adaptation investigated in this paper is thus defined as follows

**Definition 3.2** (Gradual Domain Adaptation)**.** *Given a finite time horizon $\{1, \ldots, T\}$, at each time step $t$, a data distribution $P_t$ over $\mathbb{R}^d \times \mathbb{R}^k$ is realized with the features $X \in \mathbb{R}^d$ and labels $Y \in \mathbb{R}^k$. The data distributions are gradually evolving subject to Assumption 3.1. With a given loss function $\ell(\cdot, \cdot)$ that satisfies Assumption 3.2, the goal is to obtain a classifier $h \in \mathcal{H} : \mathbb{R}^d \to \mathbb{R}^k$ that minimizes the given loss function on the target domain $P_T$ with access to only $n$ samples from each intermediate domain $P_1, \ldots, P_{T-1}$.*

Our first tool is used to help us characterize the structure of sequential domain adaptation. Under the statistical learning scenario with i.i.d. data, Rademacher complexity serves as a well-known complexity notion to capture the richness of the underlying hypothesis space. However, with the sequential dependence, classical notions of complexity are insufficient to provide a description of the problem. To capture the difficulty of sequential domain adaptation, we use the sequential Rademacher complexity, which was originally proposed for online learning where data comes one by one in sequence (Rakhlin et al., 2015).

**Definition 3.3** ($\mathcal{Z}$-valued tree Rakhlin et al. (2015))**.** *A $\mathcal{Z}$-value tree $z$ is a sequence $(z_1, \ldots, z_T)$ of $T$ mappings, $z_t : \{\pm 1\}^{t-1} \to \mathcal{Z}, t \in [1, T]$. A path in the tree is $\epsilon = (\epsilon_1, \ldots, \epsilon_{T-1}) \in \{\pm 1\}^{T-1}$. To simplify the notations, we write $z_t(\epsilon) = z_t(\epsilon_1, \ldots, \epsilon_{t-1})$.*

**Definition 3.4** (Sequential Rademacher Complexity Rakhlin et al. (2015))**.** *For a function class $\mathcal{F}$, the sequential Rademacher complexity is defined as $\mathfrak{R}_T^{\mathrm{seq}}(\mathcal{F}) = \sup_{\mathbf{z}} \mathbb{E} \left[ \sup_{f \in \mathcal{F}} \frac{1}{T} \sum_{t=1}^{T} \epsilon_t f(z_t(\epsilon)) \right]$ , where the*

*supremum is taken over all $\mathcal{Z}$-valued trees (Definition 3.3) of depth $T$ and $(\epsilon_1, \ldots, \epsilon_T)$ are Rademacher random variables.*

We next introduce the discrepancy measure, a key ingredient that helps us to characterize the non-stationarity resulting from the shifting data domains. This can be used to bridge the shift in data distribution with the shift in errors incurred by the classifier. To simplify the notation, we let $Z = (X, Y)$ and use shorthand $Z_1^T$ for $Z_1, \ldots, Z_T$.

**Definition 3.5** (Discrepancy measure Kuznetsov & Mohri (2020))**.**

$$\text{disc}_T = \sup_{h \in \mathcal{H}} \left( \mathbb{E}\left[ \ell_h\left(X_T, Y_T\right) \mid Z_1^{T-1} \right] - \frac{1}{T} \sum_{t=1}^{T} \mathbb{E}\left[ \ell_h\left(X_t, Y_t\right) \mid Z_1^{t-1} \right] \right), \tag{1}$$

*where $Z_1^0$ is defined to be the empty set, in which case the expectation is equivalent to the unconditional case.*

We will later show that the discrepancy measure can be directly upper-bounded when the shift in class conditional distribution is gradual. We also note that this notion is general and feasible to be estimated from data in practice (Kuznetsov & Mohri, 2020). Similar notions have also been used extensively in non-stationary time series analysis and mixing processes (Kuznetsov & Mohri, 2014; 2017).

## 4 Theoretical Results

In this section, we provide our theoretical guarantees for the performance of the final classifier learned in the setting described above. Our result is algorithm agnostic and general to loss functions that satisfy Assumption 3.2. We then discuss the implications of our results and give a proof sketch to illustrate the main ideas.

The following theorem gives an upper bound of the expected loss of the learned classifier on the last domain in terms of the shift $\Delta$, sequential Rademacher complexity, etc.

**Theorem 4.1.** *Under Assumptions 3.1, 3.2, with $n$ data points access to each data distribution $P_t$, $t \in \{1, \ldots, T\}$, and loss function $\ell_h = \ell(h(x), y) : \mathcal{X} \times \mathcal{Y} \to \mathbb{R}$, the loss on the last distribution incurred by a classifier $h_T$ obtained through empirically minimizing the loss on each domains can be upper bounded by*

$$\mathbb{E}\left[ \ell_{h_T}\left(X_T, Y_T\right) \mid Z_1^{T-1} \right]$$
$$\leq \mathbb{E}\left[ \ell_{h_0}\left(X_T, Y_T\right) \mid Z_1^{T-1} \right] + \underbrace{\frac{3}{T} + \frac{3M}{T}\sqrt{8 \log \frac{1}{\delta}}}_{E_1} + \underbrace{\frac{1}{T}\sqrt{\frac{VCdim(\mathcal{H}) + \log(2/\delta)}{2n}} + O\left(\frac{1}{\sqrt{nT}}\right)}_{E_2}$$
$$+ \underbrace{18M\sqrt{4\pi \log T} \, \mathfrak{R}_{T-1}^{seq}(\mathcal{F}) + 3T\rho\Delta}_{E_3}, \tag{2}$$

*where $\ell_h \in \mathcal{F}$, $\mathfrak{R}_T^{seq}(\mathcal{F})$ is the sequential Rademacher complexity of $\mathcal{F}$, $VCdim(\mathcal{H})$ is the VC dimension of $\mathcal{H}$ and $h_0 = \text{argmin}_{h \in \mathcal{H}} \frac{1}{T} \sum_{t=1}^{T} \ell\left(h(X_t), Y_t\right)$.*

When $\ell_h \in \mathcal{F}$ is bounded and convex, the sequential Rademacher complexity term is upper bounded by $O(\sqrt{1/nT})$ (Rakhlin et al., 2015). For some complicated function classes, such as multi-layer neural networks, they also enjoy a sequential Rademacher complexity of order $O(\sqrt{1/nT})$ (Rakhlin et al., 2015). Before we move to present a proof sketch of Theorem 4.1, we first discuss the implications of our theorem.

**Remark 4.1.** *There exists a non-trivial trade-off between $E_1 + E_2$ and $E_3$ through the length $T$. When $T$ is larger, all terms except for the terms in $E_3$ will be smaller while the terms in $E_3$ will be larger. Hence, it is not always beneficial to have a longer trajectory.*

**Remark 4.2.** *All terms in (2) except for the last term $3T\rho\Delta$ are determined regardless of the algorithm. The last term depends on $\Delta$ which measures the class conditional distance between any two consecutive domains. This distance can potentially be minimized through learning an effective representation of data.*

**Comparison with unsupervised gradual domain adaptation** Our result is only linear with respect to the horizon length $T$ and the average loss $\mathbb{E}\left[\ell_{h_0}\left(X_T, Y_T\right) \mid Z_1^{T-1}\right]$, where $h_0 = \operatorname{argmin}_{h \in \mathcal{H}} \frac{1}{T} \sum_{t=1}^{T} \ell\left(h(X_t), Y_t\right)$. In contrast, the previous upper bound given by Kumar et al. (2020), which is for unsupervised gradual domain adaptation, is $O(\exp(T)\alpha_0)$, with $\alpha_0$ being the initial error on the initial domain. It remains unclear, however, if the exponential cost is unavoidable when labels are missing during training as the result by Kumar et al. (2020) is algorithm specific.

**Comparison with multiple source domain adaptation** The setting of multiple source domain adaptation neglects the temporal structure between training domains. Our results are comparable while dropping the requirement of simultaneous access to all training domains. Our result suffers from the same order of error with respect to the Rademacher complexity and from the VC inequality with supervised multiple source domain adaptation (MDA) (Wen et al., 2020), which is $O\left(\sum_t \alpha_t(\ell_{h_T}(X_t, Y_t) + \mathfrak{R}(\mathcal{F}))\right)$, where $\sum_t \alpha_t = 1, \alpha_t > 0, \forall t$ and $\mathfrak{R}(\mathcal{F})$ being the Rademacher complexity. Taking the weights $\alpha_t = \frac{1}{T}$, the error of a classifier $h$ on the target domain similarly relies on the average error of $h$ on training domains. We note that in comparison our results scale with the averaged error of the best classifier on the training domains.

While we defer the full proof to the appendix, we now present a sketch of the proof.

*Proof Sketch* With Assumption 3.1, we first show that when the Wasserstein distance between two consecutive class conditional distributions is bounded, the discrepancy measure is also bounded.

**Lemma 4.1.** *Under Assumption 3.2, the expected loss on two consecutive domains satisfy* $\mathbb{E}_\mu[\ell_h(X, Y)] - \mathbb{E}_\nu[\ell_h(X', Y')] \leq \rho\Delta$ *, where* $\mu, \nu$ *are the probability measure for* $P_t, P_{t+1}$, $(X, Y) \sim P_t$, *and* $(X', Y') \sim P_{t+1}$.

Then we leverage this result to bound the loss incurred in expectation by the same classifier on two consecutive data distributions. We start by decomposing the discrepancy measure with an adjustable summation term as

$$\operatorname{disc}_T \leq \sup_{h \in \mathcal{H}} \left(\frac{1}{s} \sum_{t=T-s+1}^{T} \mathbb{E}\left[\ell_h\left(X_t, Y_t\right) \mid Z_1^{t-1}\right] - \frac{1}{T} \sum_{t=1}^{T} \mathbb{E}\left[\ell_h\left(X_t, Y_t\right) \mid Z_1^{t-1}\right]\right)$$
$$+ \sup_{h \in \mathcal{H}} \left(\mathbb{E}\left[\ell_h\left(X_T, Y_T\right) \mid Z_1^{T-1}\right] - \frac{1}{s} \sum_{t=T-s+1}^{T} \mathbb{E}\left[\ell_h\left(X_t, Y_t\right) \mid Z_1^{t-1}\right]\right).$$

We show by manipulating this adjustable summation, the discrepancy measure can indeed be directly obtained through an application of Lemma 4.1. We now start to bound the learning error in interest by decomposing

$$\mathbb{E}\left[\ell_{h_T}\left(X_T, Y_T\right) \mid Z_1^{T-1}\right] - \mathbb{E}\left[\ell_{h_0}\left(X_T, Y_T\right) \mid Z_1^{T-1}\right] \leq 2\Phi(Z_1^T) + \left(\frac{1}{T} \sum_{t=1}^{T-1}\left[\ell_{h_T}\left(X_t, Y_t\right)\right] - \frac{1}{T} \sum_{t=1}^{T-1} \ell_{h_0}\left(X_T, Y_T\right)\right),$$

where $\Phi\left(Z_1^T\right) = \sup_{h \in \mathcal{H}}\left(\mathbb{E}\left[\ell_h\left(X_T, Y_T\right) \mid Z_1^{T-1}\right] - \sum_{t=1}^{T} \frac{1}{T} \ell_h\left(X_t, Y_t\right)\right)$. The term $\Phi\left(Z_1^T\right)$ can be upper bounded by Lemma B.1 Kuznetsov & Mohri (2020) and thus it is left to bound the remaining term $\frac{1}{T} \sum_{t=1}^{T-1}\left[\ell_{h_T}\left(X_t, Y_t\right)\right] - \frac{1}{T} \sum_{t=1}^{T-1} \ell_{h_0}\left(X_T, Y_T\right)$. To upper bound this difference of average loss, we first compare the loss incurred by a classifier learned by an optimal online learning algorithm to $f_0$. By classic online learning theory results, the difference is upper bounded by $O\left(\frac{1}{\sqrt{nT}}\right)$. Then we compare the optimal online learning classifier to our final classifier $h_T$ and upper bound the difference through the VC inequality (Bousquet et al., 2004).

Lastly, we leverage Corollary 3 of Kuznetsov & Mohri (2020) with our terms to complete the proof. $\square$

## 5 Insights for Practice

The key insight indicated by Theorem 4.1 and Remark 4.2 is that the bottleneck of supervised gradual domain adaption is not only predetermined through the setup of the problem but also relies heavily on $\rho\Delta$, where $\Delta$ is the upper bound of the Wasserstein class conditional distance between two data domains and $\rho$ is the Lipschitz constant of the loss function. In practice, the loss function is often chosen beforehand and

remains unchanged throughout the learning process. Therefore, the only term available to be optimized is $\Delta$, which can be effectively reduced if a good representation of data can be learned for classification. We give a feasible primal-dual objective that learns a mapping function from input to feature space concurrently with the original classification objective

**A primal-dual objective formulation**  Define $g$ to be a mapping that maps $X \in \mathbb{R}^d$ to some feature space. We propose the learning objective as to learn a classifier $h$ simultaneously with the mapping function $g$ with the exposure of historical data $Z_1^{T-1}$. With the feature $g(X)$ from the target domain, our learning objective is now $\mathbb{E}\left[\ell_h(g(X_T), Y_T))|Z_1^{T-1}\right] - \inf_{h^*, g^*} \mathbb{E}\left[\ell_{h^*}(g^*(X_T), Y_T)|Z_1^{T-1}\right]$. Intuitively, this can be viewed as a combination of two optimization problems where both $\Delta$ and the learning loss are minimized.

The objective is hard to evaluate without further assumptions. Thus we restrict our study to the case where both $g$ and $h$ are parametrizable. Specifically, we assume $g$ is parameterized by $\omega$ and $h$ is parameterized by $\theta$. Then we leverage the Wasserstein-1 distance's dual representation (Equation 3, Kantorovich & Rubinstein (1958)) to derive an objective for learning the representation,

$$W_1(P, Q) = \sup_{\gamma \in \Gamma(P,Q)} \int \gamma(x)dP(x) - \int \gamma(y)dQ(y) = \sup_{\gamma \in \Gamma(P,Q)} \mathbb{E}_P[\gamma(x)] - \mathbb{E}_Q[\gamma(y)]. \tag{3}$$

With this, the following primal-dual objective can be used to concurrently find the best-performing classifier and representation mapping,

$$\min_\theta \max_\omega \mathbb{E}\left[\ell_{h_{\theta,T}}\left(g_\omega(X_T), Y_T\right) \mid Z_1^{T-1}\right] + \lambda L_D, \tag{4}$$

where $L_D = \max_t \mathbb{E}_{P_t}\left[g_\omega(X_t)\right] - \mathbb{E}_{P_{t+1}}\left[g_\omega(X_{t+1})\right]$ and $\lambda$ is a tunable parameter.

**One-step and temporal variants**  Notice that $L_D$ relies on the maximum distance across all domains. It is thus hard to directly evaluate $L_D$ without simultaneous access to all domains. With access only to the current and the past domains, we could optimize the following one-step primal-dual loss at time $t$ instead.

$$\min_\theta \max_\omega \mathbb{E}\left[\ell_{h_{\theta,t}}\left(g_\omega(X_t), Y_t\right) \mid Z_1^t\right] + \lambda L_{D_t}, \tag{5}$$

where $L_{D_t} = \mathbb{E}_{P_t}\left[g_\omega(X_t)\right] - \mathbb{E}_{P_{t+1}}\left[g_\omega(X_{t+1})\right]$.

Compared to the objective (4), the one-step loss (5) only gives us partial information, and directly optimizing it may often lead to suboptimal performance. While it is inevitable to optimize with some loss of information under the problem setup, we use a temporal model (like an LSTM) to help preserve historical data information in the process of learning mapping function $g$. In particular, in the temporal variant, we will be using the hidden states of an LSTM to dynamically summarize the features from all the past domains. Then, we shall use the feature distribution computed from the LSTM hidden state to align with the feature distribution at the current time step.

To practically implement these objectives, we can use neural networks to learn the representation and the classifier, and another neural network is used as a critic to judge the quality of the learned representations. To minimize the distance between representations of different domains, one can use $W_1$ distance as an empirical metric. Then the distance of the critic of the representations in different domains is then minimized to encourage the learning of similar representations. We note that the use of $W_1$ distance, which is easy to evaluate empirically, to guide representation learning has been practiced before (Shen et al., 2018). We take this approach further to the problem of gradual domain adaption.

## 6  Empirical Results

In this section, we perform experiments to demonstrate the effectiveness of supervised gradual domain adaptation and compare our algorithm with No Adaptation, Direct Adaptation, and Multiple Source Domain Adaptation (MDA) on different datasets. We also verify the insights we obtained in the previous section by answering the following three questions:

1. **How helpful is representation learning in gradual domain adaptation?** Theoretically, effective representation where the data drift is "small" helps algorithms to gradually adapt to the evolving domains. This corresponds to minimizing the $\rho\Delta$ term in our Theorem 4.1. We show that our algorithm with objective (5) outperforms the objective of empirical risk (No Adaptation).

2. **Can the one-step primal-dual loss (5) act as an substitute to optimization objective (4)?** Inspired by our theoretical results (Theorem 4.1), the primal-dual optimization objective (4) should guide the adaptation process. However, optimization of this objective requires simultaneous access to all data domains. We use a temporal encoding (through a temporal model such as LSTM) of historical data to demonstrate the importance of the information of past data domains. We compare this to results obtained with a convolutional network (CNN)-based model to verify that optimizing the one-step loss (5) with a temporal model could largely mitigate the information loss.

3. **Does the length of gradual domain adaptation affect the model's ability to adapt?** Our theoretical results suggest that there exists an optimal length $T$ for gradual domain adaptation. Our empirical results corroborate this as when the time horizon passes a certain threshold the model performance is saturated.

## 6.1 Experimental Setting

We conduct our experiments on Rotating MNIST, Portraits, and FMOW, with a detailed description of each dataset in the appendix. We compare the performance of no adaptation, direct adaptation, and multiple source domain adaptations with gradual adaptation. The implementation of each method is also included in the appendix. Each experiment is repeated over 5 random seeds and reported with the mean and 1 std.

## 6.2 Experimental Results

Table 1: Results on rotating MNIST dataset with Gradual Adaptation on 5 domains, Direct Adaptation, and No Adaptation.

| Rotating MNIST with 5 domains | | | | | |
|---|---|---|---|---|---|
| | Gradual Adaptation | | Direct Adaptation | | No Adaptation |
| | CNN | LSTM | CNN | LSTM | CNN |
| 0-30 degree | $90.21 \pm 0.48$ | $\mathbf{94.83} \pm 0.49$ | $77.97 \pm 0.99$ | $89.72 \pm 0.73$ | $79.76 \pm 3.20$ |
| 0-60 degree | $87.35 \pm 1.02$ | $\mathbf{92.52} \pm 0.25$ | $73.27 \pm 1.51$ | $88.53 \pm 0.76$ | $58.36 \pm 2.59$ |
| 0-120 degree | $82.38 \pm 0.57$ | $\mathbf{89.72} \pm 0.35$ | $62.52 \pm 1.06$ | $84.30 \pm 2.60$ | $38.25 \pm 0.61$ |

Table 2: Results on rotating MNIST dataset (5 domains) with Gradual Adaptation, MDA (MDAN) (Zhao et al., 2018) and Meta-learning (EAML) (Liu et al., 2020).

| Rotating MNIST with 5 domains | | | | | | |
|---|---|---|---|---|---|---|
| | Gradual Adaptation | | MDAN | | | EAML |
| | CNN | LSTM | Maxmin | Dynamic | Dynamic with last 2 domains | CNN |
| 0-30 degree | $90.21 \pm 0.48$ | $9.83 \pm 0.49$ | $93.62 \pm 0.87$ | $\mathbf{95.79} \pm \mathbf{0.33}$ | $83.04 \pm 0.29$ | $79.10 \pm 6.99$ |
| 0-60 degree | $87.35 \pm 1.02$ | $\mathbf{92.52} \pm \mathbf{0.25}$ | $91.99 \pm 0.51$ | $92.27 \pm 0.26$ | $61.49 \pm 0.72$ | $54.64 \pm 4.08$ |
| 0-120 degree | $82.38 \pm 0.57$ | $\mathbf{89.72} \pm \mathbf{0.35}$ | $87.25 \pm 0.52$ | $88.57 \pm 0.21$ | $41.14 \pm 1.77$ | $17.91 \pm 10.34$ |

Table 3: Results on FMOW with Gradual Adaptation with 3 domains, Direct Adaptation, and No Adaptation.

| FMOW | | | |
|---|---|---|---|
| No Adaptation with ERM | Direct Adaptation with CNN | Gradual Adaptation with CNN | Gradual Adaptation with LSTM |
| $33.10 \pm 1.94$ | $41.94 \pm 2.73$ | $36.86 \pm 1.91$ | $\mathbf{43.52} \pm 1.40$ |

**Learning representations further help in gradual adaptation**

On rotating MNIST, the performance of the model is better in most cases when adaptation is considered (Table 1), which demonstrates the benefit of learning proper representations. With a CNN architecture, the only exception is when the shift in the domain is relatively small (0 to 30 degree), where the No Adaptation method achieves higher accuracy than the Direct Adaptation method by 2%. However, when the shift in domains is relatively large, Adaptation methods are shown to be more successful in this case and this subtle advantage of No Adaptation no longer holds. Furthermore, Gradual Adaptation further enhances this outperformance significantly. This observation shows the advantage of sequential adaptation versus direct adaptation. We further show that the performance of the algorithm monotonically increases as it progress

Table 4: Results on Portraits with Gradual Adaptation for different lengths of horizon $T$, Direct Adaptation, and No Adaptation.

| Portraits | | |
|---|---|---|
| | CNN | LSTM |
| No Adaptation | $76.01 \pm 1.45$ | N/A |
| Direct Adaptation | $86.86 \pm 0.84$ | N/A |
| Gradual - 5 Domains | $87.77 \pm 0.98$ | $87.41 \pm 0.76$ |
| Gradual - 7 Domains | $89.14 \pm 1.64$ | $89.15 \pm 1.12$ |
| Gradual - 9 Domains | $90.46 \pm 0.54$ | $89.88 \pm 0.54$ |
| Gradual - 11 Domains | $90.56 \pm 1.21$ | $90.93 \pm 0.75$ |
| Gradual - 12 Domains | $91.45 \pm 0.27$ | $90.73 \pm 0.66$ |
| Gradual - 13 Domains | $90.54 \pm 0.90$ | $91.13 \pm 0.35$ |
| Gradual - 14 Domains | $90.58 \pm 0.38$ | $90.35 \pm 0.71$ |

to adapt to each domain and learn a cross-domain representation. Figure 1b shows the trend in algorithm performance on rotating MNIST and FMOW.

**One-step loss is insufficient as a substitute, but can be improved by temporal model**   The inefficiency of adaptation without historical information appears with all datasets we have considered, reflected through Table 1, 3, 4. In almost all cases, we observe that learning with a temporal model (LSTM) achieves better accuracy than a convolutional model (CNN). The gap is especially large on FMOW, the large-scale dataset in our experiments. We suspect that optimizing with only partial information can lead to suboptimal performance on such a complicated task. This is reflected through the better performance achieved by Direct Adaptation with CNN when compared to Gradual Adaptation with CNN and 3 domains (Table 3). In contrast, Gradual Adaptation with LSTM overtakes the performance of Direct Adaptation, suggesting the importance of historical representation. Another evidence is that Figure 1b shows that Gradual Adaptation with a temporal model performs better on all indexes of domains on rotating MNIST and FMOW.

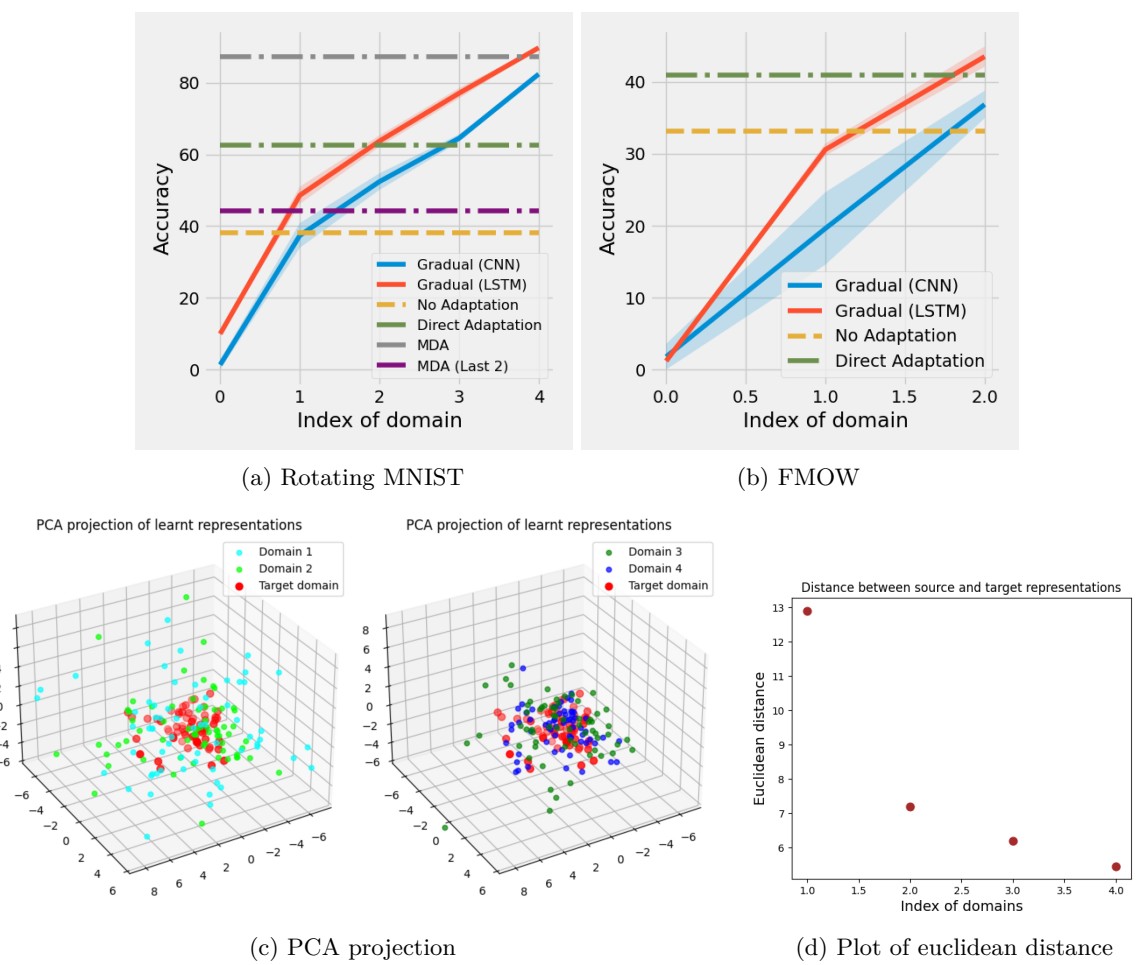

(a) Rotating MNIST

(b) FMOW

(c) PCA projection

(d) Plot of euclidean distance

Figure 1: Figure 1a compares the training curves on rotating MNIST with maximum rotation of 120 degrees. Figure 1b compares the training curves on FMOW. Figure 1c is the PCA projection plot of learned representation and Figure 1d plots the Euclidean distance to the target domain of the projections of learned representations.

**Existence of optimal time horizon**     With the Portraits dataset and different lengths of horizon $T$, we verify that then optimal time horizon can be reached when model performance is saturated in Table 4. The performance of the model increases drastically when the shifts in domains are considered, shown by the difference in the performance of No Adaptation, Direct Adaptation, and Gradual Adaptation with 5 and 7 domains. However, this increase in performance becomes relatively negligible when $T$ is large (the performance gain is saturated when the horizon length is $9 - 14$). This rate of growth in accuracy implies that there exists an optimal number of domains.

**Comparison with MDA**     Lastly, we remark on the results (Table 2 and 5) achieved by Gradual Adaptation in comparison with MDA methods (MDAN (Zhao et al., 2018), DARN (Wen et al., 2020) and Fish (Shi et al., 2022)). On Rotating MNIST, we note that Gradual Adaptation outperforms MDA methods when the shift is large (60 and 120 degree rotation) while relaxing the requirement of simultaneous access to all source domains. It is only when the shift is relatively small (30-degree rotation), MDA method DARN achieves a better result than ours. When the MDA method is only presented with the last two training domains, Gradual Adaptation offers noticeable advantages regardless of the shift in the domains (Table 2). This demonstrates the potential of graduate domain adaptation in real applications that even when the data are not simultaneously presented it is possible to achieve competitive or even better performance. One possible reason for this can be illustrated by Figure 1d, in which we plot the PCA projections and the

Table 5: Results on rotating MNIST dataset with Gradual Adaptation on 5 domains and MDA methods, Fish (Shi et al., 2022) and DARN (Wen et al., 2020)

|  | Fish | DARN | Ours |
|---|---|---|---|
| 0-30 degree | **95.83 ± 0.13** | 94.20 ± 0.27 | 94.83 ± 0.49 |
| 0-60 degree | 90.57 ± 0.37 | 89.50 ± 0.12 | **92.52 ± 0.25** |
| 0-120 degree | 83.26 ± 1.58 | 82.28 ± 2.42 | **89.72 ± 0.35** |

Euclidean distance to the target domain of learned representations. From Figure 1d, we can see that the gradual domain adaptation method is able to gradually learn an increasingly closer representation of the source domain to the target domain. This helps our method to make our prediction based on more relevant features while MDA methods may be hindered by not-so-relevant features from multiple domains.

**Comparison with Meta-learning**  In addition to comparing our method with those achieved by domain adaptation methods, we also compare our work with Meta-learning for evolving distributions (EAML (Liu et al., 2020)). In contrast to gradual domain adaptation, these methods instead assume that the testing distribution is gradually evolving and thus the learned classifier is desired to be adaptive to these changing targets. In addition, the learned classifier is also asked to "not forget". Due to the different objectives, we can see that the baseline method (EAML) is suboptimal on a fixed target with changing training distributions, when compared to our method (Table 2).

## 7  Conclusion

We studied the problem of supervised gradual domain adaptation, which arises naturally in applications with temporal nature. In this setting, we provide the first learning bound of the problem and our results are general to a range of loss functions and are algorithm agnostic. Based on the theoretical insight offered by our theorem, we designed a primal-dual learning objective to learn an effective representation across domains while learning a classifier. We analyze the implications of our results through experiments on a wide range of datasets.

## Acknowledgements

We want to thank the reviewers and the editors for their constructive comments during the review process. Jing Dong and Baoxiang Wang are partially supported by National Natural Science Foundation of China (62106213, 72150002) and Shenzhen Science and Technology Program (RCBS20210609104356063, JCYJ20210324120011032). Han Zhao would like to thank the support from a Facebook research award.

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
