# OpenReview forum: "Algorithms and Theory for Supervised Gradual Domain Adaptation"
_TMLR — Accepted by TMLR_

### Review · Reviewer_ybbK · 2022-09-14

**Summary Of Contributions:**

This paper considers a supervised version of gradual domain adaptation---which has gathered attention in recent years due to intrinsic limitations of the usual (one-shot) domain adaptation. In this scenario, a learner shall be exposed to training data drawn from a joint probability distribution, which will be drifting over time gradually in the sense that the class-conditional distributions shift without moving too far from the previous distribution in terms of the Wasserstein distance. Under this setup, the authors derive the target error bound (by leveraging the sequential Rademacher complexity and discrepancy measure), which the authors claim improves the previous bound obtained by Kumar et al. (2020) regarding the dependency on the initial error. Inspired by the derived bound, the authors propose a min-max-based representation learning algorithm to minimize the distance between consecutive distributions. Additionally proposed temporal encoding with the LSTM improves the data efficiency, without access to data from the past domains. Finally, the authors confirm that the proposed method experimentally outperforms domain adaptation methods without leveraging multiple domains and multi-source domain adaptation (which discards the temporal structure).

**Broader Impact Concerns:**

This does not apply to the current submission.

**Requested Changes:**

Major comments
====

The followings were mentioned above.

- To make the literature review self-contained, in particular Kumar et al. (2020).
- To check the monotonicity of the $p$-Wasserstein distance carefully.
- To make the algorithm derivation clearer.

Some minor comments
====

- The related work section is usually written in the past tense (or perfect tense). In Section 2, I found several sentences in the present tense.
- In Assumption 3.1, the initial $1 \le t \le T$ seems to be $1 \le t \le T - 1$, otherwise the undefined $P_{T+1}$ will appear.
- In Assumption 3.2, "through out" -> "throughout".
- In Assumption 3.3, when you mention the Lipschitz continuity (of the bivariate loss function), it would be better to clearly say "Lipschitz continuous with respect to $(x,y)$".
- Right after Assumption 3.3, "In the case where the input data fails to be compact, the assumption remains true after the normalization of data", which I don't think is accurate because the radius of the unbounded real vector space (not compact) is the infinity and hence the normalization makes all points to be the origin.
- In Definition 2, $\mathcal{Z}$-valued trees should be defined properly to make them self-contained.
- In Definition 3, please check the superscript $t-1$ in the condition of the second conditional expectation. If we take the sum from $t=1$ to $T-1$, $Z^0$ will appear, which seems strange.
- In Theorem 4.1, Assumption 3.2 should be assumed as well.
- In Theorem 4.1, it is not clear how "a learned classifier $h\_T$" is obtained/defined. I guess this should be "for any $h\_T$".
- In Lemma 4.1, please remove the period after "the expected loss on two consecutive domains satisfy".
- In the last equation of the page 5, the summation $\sum\_{t=1}^T$ is not consistent to the definition of $\mathrm{disc}\_T$ in equation (1). Could you check it again?
- In Section 5, it would be better to cite "the Wasserstein-1 distance's dual representation".
- After equation (4), the second expectation should be $\mathbb{E}\_{P\_{t+1}}$ instead of $\mathbb{E}\_{P\_{t-1}}$.
- In the last paragraph of Section 5, I don't understand where we need "to approximate the Wasserstein distance".
- In Section 6, to verify research question 2 "Can the one-step primal-dual loss act as a substitute to optimization objective (3)?", I would encourage the authors not only to compare the LSTM and CNN but also to compare the original objective (3) by using all past domains.
- In Table 4, although the performance seems to saturate around $T=9$ or $T=11$, we cannot be sure that they are optimal time horizons. Is it possible to test larger horizons to see if they are indeed optimal?
- When we look at Figure 1a, the multi-source domain adaptation baseline seems to work very competitively with the proposed method, but this baseline is not tested in Figure 1b and Table 3. Why don't you compare it to see how the baseline performs in these settings?
- In Appendix A, "lemma 4.1" in the section title should be capitalized.
- In reference, "David" should be corrected to "Ben-David".

**Strengths And Weaknesses:**

Strengths
====

1. **Practical importance of problem setting**.
Despite a long history of research on (unsupervised) domain adaptation, they often face the challenge of how to overcome vastly different source and target domains.
This challenge is reflected in many well-known domain adaptation error bounds such as Ben-David et al. (2007) and Mansour et al. (2009).
Indeed, adapting from a source domain vastly different from a target domain is not a well-defined problem and is often too pessimistic.
Thus, it is more natural to consider gradual domain adaptation in real-world scenarios.
This paper takes one step further to make it supervised, which is naturally arising in real-world scenarios and benefits a learner to learn in a better way.
The authors motivate this perspective well in the introduction.

2. **Improvement of the error bound**.
As one of the benefits to consider the supervised problem, the authors show that the domain adaptation error bound they derived depends on the initial domain error linearly.
As they claim, this is in stark contrast to the error bound derived by Kumar et al. (2020) in unsupervised gradual domain adaptation, which depends exponentially.
This draws an interesting insight to consider the supervised problem.
(But, I was not fully convinced of this perspective because I could not validate the authors' claim regarding Kumar et al.'s results. See Weaknesses #1.)

Weaknesses
====

1. **Comparison/review of existing studies may not be sufficient/self-contained.**
Some parts of the paper regarding the discussion of previous studies are not self-contained so the readers may feel difficult to validate the correctness of the paper.
The most significant point is the discussion of Kumar et al. (2020).
Although the authors claim "the given upper bound of the learning error on the target domain suffers from exponential dependency (in terms of the length of the trajectory) on the initial learning error on the source domain" in the introduction,
I did not understand which claim made by Kumar et al. (2020) corresponds to it.
Their main theorem (and its corollary) Corollary 3.3 shows the exponential dependency $\beta^{T+1}$, where $\beta = \frac{2}{1-\rho R}$ ($\rho$ is a kind of the Lipschitz constant and $R$ is the maximal radius of the hypothesis set).
There does not seem to be any exponential dependency on the initial learning error as far as I understood.
The authors may clarify this part in the paper to convince the readers of the contribution.
Moreover, since Kumar et al. (2020) is one of the closely related previous studies, the authors may consider reviewing this work in more detail (by using equations).

In addition, some other literature reviews can be made more informative.
I felt the following reviews in Section 2 are insufficient to draw connections to the current work.

- "The theoretical limits of domain adaptations have also been extensively studied (David et al., 2010; Zhao et al., 2019a; Wu et al., 2019a; Zhao et al., 2020)."  // What are the theoretical limitations? How do these works differ?
- "This result is later ... improved by concurrent independent work Wang et al., (2022)."  // Even if it is concurrent, it is important to provide a sufficient review to show what is different from the current submission.

2. **(Probably) incorrect arguments regarding monotonicity of the Wasserstein distance**.
In Section 3, the authors claim that "due to the monotonicity of the $W\_p$ distance, the $W\_1$ distance leads to tighter upper bounds", and use the inequality $W\_1(\mu, \nu) \le W\_p(\mu, \nu)$ in the proof of Lemma 3.1.
Although I did not find a nice proof immediately, I don't think this is correct.
The intuition comes from the monotonicity of $\ell\_p$ norms: $\ell\_p$ norm decreases as $p$ gets larger.
Since the $p$-Wasserstein distance is defined by using the $\ell\_p$ norm as the base metric (although it involves the minimization), the analogy seems to work here.
I did a simple simulation below, which shows an example that the aforementioned inequality may be incorrect.
(`ot` is the Python OT package)
```
In [1]: import numpy as np
   ...: import ot
   ...: from ot.datasets import make_1D_gauss as gauss

In [2]: n = 100  # nb bins
   ...:
   ...: # bin positions
   ...: x = np.arange(n, dtype=np.float64)
   ...:
   ...: # Gaussian distributions
   ...: a = gauss(n, m=20, s=5)  # m= mean, s= std
   ...: b = gauss(n, m=60, s=10)
   ...:
   ...: # loss matrix
   ...: M = ot.dist(x.reshape((n, 1)), x.reshape((n, 1)))
   ...: M /= M.max()

In [3]: for p in range(1, 10):
   ...:     w = ot.wasserstein_1d(a, b, p=p)
   ...:     print(f"{p} {w}")
   ...:
1 0.0064531459404555715
2 0.0001326386522362971
3 3.716677655553046e-06
4 1.20986120423101e-07
5 4.229773302944304e-09
6 1.531836302744079e-10
7 5.6554649157605976e-12
8 2.112946713019602e-13
9 7.959239997625665e-15
```
I suppose that the proof of Lemma 4.1 could be corrected by assuming the Lipschitz continuity of the loss function with respect to the $\ell\_p$ norm (instead of $\ell\_1$ norm in the current form) and correcting Assumption 3.1 from $W\_p \le \Delta$ to $W\_p^p \le \Delta$, which still requires careful investigations.
Due to this (probably) inaccurate claim, we may need to rethink the comparison of the assumptions with Kumar et al. (2020).

3. **The provided theory assumes the class-conditional shift, too**.
Although the authors criticize Kumar et al. (2020) by "They assumed that the label distribution remains unchanged while the varying class conditional probability ... which only covers a limited number of cases." in Section 2,
the authors eventually assume the same one in Assumption 3.2.
I think it is okay to assume it for the time being, but this argument may need clarification.

4. **The algorithmic formulation has remained not sufficiently clear**.
In Section 5, the authors provide a practical algorithm based on the error bound to learn representation by making $\Delta$ smaller.
However, the presentation of the algorithm derivation may not be sufficiently clear because of the following points.

- At the beginning of the second paragraph of "A primal-dual objective formulation", we see "the objective (3)", which has not appeared at the moment of this sentence. Equation (3) appears subsequently, which does not seem to be what is mentioned here in light of the context.
- The authors mention "the Wasserstein-1 distance's dual representation", but it is not clear on which part (in the error bound?) the Kantrovich duality has been applied. Accordingly, equation (3) could not be interpreted.

---

> ### Author Response · Authors · 2022-10-08
> **Response to reviewer ybbK (part 1: comparison with previous results)**
>
> We thank the reviewer for the detailed and constructive comments. We have fixed the typos and clarified the notations pointed out by the reviewer. We will address the main concerns raised in the following section.
>
> ### 1. Detailed comparison with previous results.
> Following the reviewer's suggestions, we have updated our related work sections to provide an in-depth comparison with detailed technical discussion to contrast existing results with ours. Specifically, when compared to previous results on GDA, our result is only linear with respect to the horizon length $T$ and the average loss $\mathbb{E}\left[\ell_{h_0}\left(X_T, Y_T\right) \mid Z_{1}^{T-1}\right]$, where $h_{0}=\operatorname{argmin}_{h \in \mathcal{H}} \frac{1}{T}\sum\_{t=1}^{T} \ell\left(h(X\_t), Y\_{t}\right)$. In contrast, the previous upper bound given by (Kumar et al. 2020), which is for unsupervised gradual domain adaptation, is $O(\alpha_0\cdot\exp(T))$, with $\alpha_0$ being the initial error on the initial domain. In terms of the dependency on the time horizon $T$, ours is an exponential improvement. This result is later improved by concurrent independent work (Wang et al. 2022) to $O(\alpha_0 + T)$ in the setting of unsupervised gradual domain adaptation.
>
> For the results obtained by (Ben-David et al. 2010; Zhao et al. 2019a; Wu et al. 2019a; Zhao et al. 2020), we have also added more details to summarize their main contributions and differences. On the theoretical side of domain adaptation, Ben-David et al., 2010 investigated the necessary assumptions for domain adaptations, which specifies that either the domains are needed to be similar, or there exists a classifier in the hypothesis class that can attain low error on both domains. For the restriction on the similarity of domains, Wu et al. 2019a proposed an asymmetrically-relaxed distribution alignment for an alternative requirement. This is a more general condition for domain adaptation when compared to those proposed by Ben-David et al., 2010, and holds in a more general setting with high-capacity hypothesis classes such as neural networks. Zhao et al. 2019a then focused on the efficiency of representation learning for domain adaption and characterized a trade-off between learning an invariant representation across domains and achieving small errors on all domains. This is then extended to a more general setting by Zhao et al. 2020, which provides a geometric characterization of feasible regions in the information plane for invariant representation learning.

---

> ### Author Response · Authors · 2022-10-08
> **Response to reviewer ybbK (part 2)**
>
> ### 2. Monotonicity of $p$-th Wasserstein distance
>
> We kindly point out that the misconception might come from the confusion between the $\ell_p$ norm and the $L_p$ norm (used in the definition of Wasserstein distance, based on probability measure). Note that these two norms are quite different. Indeed, for any fixed vector $x\in\mathbb{R}^d$, we know that $\\|x\\|\_p \geq \\|x\\|\_q$ for $1 \leq p \leq q$. However, for any probability measure $\mu$ and a function $f\in L_q(\mu)$, we have $\\|f\\|\_{L_p(\mu)}\leq \\|f\\|_{L_q(\mu)}$ for $1\leq p\leq q$.
>
> More specifically, we can rigorously prove the above inequality, i.e., the monotonicity of the Wasserstein distance as follows.  First, due to Jensen's inequality, let $\gamma$ be any coupling between $\mu$ and $\nu$, and recall that $f(t) := |t|^{q/p}$ is convex in $t$, we have
> \begin{align}
>     \left( \int \\| x - y\\|^p d \gamma \right)^{1/p} &= \left(\mathbb{E}\_\gamma[\\|x - y\\|^p]\right)^{1/p}
>     = f\left(\mathbb{E}\_\gamma[\\|x - y\\|^p]\right)^{1/q} ,
> \end{align}
> and
> $$ f\left(\mathbb{E}\_\gamma[\\|x - y\\|^p]\right)^{1/q}
>     \leq \left(\mathbb{E}\_\gamma[f\left(\\|x - y\\|^p]\right)\right)^{1/q}  = \left(\mathbb{E}\_\gamma[\\|x - y\\|^q]\right)^{1/q}
>     = \left( \int \\| x - y\\|^q d \gamma \right)^{1/q} ,$$
> where $x \sim \mu, y \sim \nu$. Since the above inequality holds for every coupling $\gamma\in\Gamma(\mu, \nu)$, it follows that
> \begin{equation*}
>     W_p(\mu,\nu) = \inf_{\gamma\in\Gamma(\mu,\nu)}\left( \int \\| x - y\\|^p d \gamma \right)^{1/p}\leq \inf_{\gamma\in\Gamma(\mu,\nu)}\left( \int \\| x - y\\|^q d \gamma \right)^{1/q} = W_q(\mu,\nu),
> \end{equation*}
> which implies the monotonicity of $W_p(\cdot,\cdot)$.
> The above proof is also included in Appendix A (proof to Lemma 4.1)
>
> We also thank the reviewer for running a hands-on experiment to validate our result. However, we'd like to kindly point out that there is a bug in the code provided by the reviewer. More specifically, the results shown by the provided code is not computing the $W_p(\mu,\nu)$ between two distributions, but instead $W_p^p(\mu,\nu)$. Please see the official doc from Python OT package regarding the ot.wassersteind\_1d function: https://pythonot.github.io/_modules/ot/lp/solver_1d.html#wasserstein_1d .
>
> In fact, after correcting the bug, the simulation results confirm our theoretical claim:
> ```
> import numpy as np
> import ot
> from ot.datasets import make_1D_gauss as gauss
>
> n = 100  # nb bins
> # bin positions
> x = np.arange(n, dtype=np.float64)
> # Gaussian distributions
> a = gauss(n, m=20, s=5)  # m= mean, s= std
> b = gauss(n, m=60, s=10)
> # loss matrix
> M = ot.dist(x.reshape((n, 1)), x.reshape((n, 1)))
> M /= M.max()
>
> for p in range(1, 10):
>    w = ot.wasserstein_1d(a, b, p=p)
>    w = w ** (1/p)
>    print(f"{p} {w}")
> ```
> Results.
> ```
> 1 0.0064531459404555715
> 2 0.011516885526751452
> 3 0.015490007649895854
> 4 0.01865021693056946
> 5 0.021147713510135707
> 6 0.02313140750373487
> 7 0.024729182459575998
> 8 0.026038206569424405
> 9 0.027128774636424272
> ```
>
> ### 3. Relaxation on class conditional shift assumption
> We have relaxed the class conditional shift assumption to include the case where the feature and the label distribution ``jointly'' shift. This is done with only some small changes to the analysis of Lemma 4.1 (in Appendix A).
>
> ### 4. Compact input data assumption
> Following the suggestion, we have removed the second part of the claim.
>
> ### 5. Optimal horizon length
> We thank the reviewer for the constructive advice. We have extended the experiments on the Portraits dataset to the cases with $12, 13, 14$ domains in the updated Table 4. We observe that, though the performance of our methods fluctuates when the horizon length is $11-14$, it is roughly saturated after the length reached $11$.
>
> ### 6. More comparisons with MDA and the original objective
> One notable difference between our method and multiple source domain adaptation (MDA) is that the latter one has simultaneous access to all the data domains. The requirement of simultaneous access implies more computational and storage overhead. On a larger dataset, such as FMOW (about 3.5TB), we are unfortunately unable to evaluate MDA due to the huge space requirements.

---

> ### Author Response · Authors · 2022-10-08
> **Response to reviewer ybbK (part 3)**
>
> ### 7. More details on the algorithm's derivation and on the need of approximating the Wasserstein distance.
> We have added more details on the derivation of the primal-dual objective. To start with, our learning objective is
> $\mathbb{E}\left[\ell\_h(g(X\_{T}), Y\_T))|Z\_1^{T-1} \right] -  \inf\_{h^\ast,g^\ast} \mathbb{E}\left[\ell\_{h^\ast} (g^\ast(X\_{T}), Y\_T)|Z\_1^{T-1} \right] $.
>
> Then we leverage the Wasserstein-$1$ distance's dual representation (the following equation) to derive an objective for learning the representation,
> $$
>     W_1 (P, Q) = \sup\_{\gamma : \\|\gamma\\|\_{\text{Lip}} \leq 1} \int \gamma(x) d P(x) - \int \gamma(y) dQ(y) = \sup\_{\gamma : \\|\gamma\\|\_{\text{Lip}} \leq 1} \mathbb{E}\_P [ \gamma] - \mathbb{E}\_Q [ \gamma] ,
> $$
> where $\\|\gamma\\|_{\text{Lip}} $ denotes the Lipschitz constant of $\gamma$.
>
> With this, the following primal-dual objective can be used to concurrently find the best-performing classifier and representation mapping,
> $$
>     \min_\theta \max_\omega \mathbb{E}\left[\ell_{h_{\theta,T}}\left(g_\omega(X_{T}), Y_T\right) \mid Z_{1}^{T-1}\right] + \lambda L_D ,
> $$
> where $L_D = \max\_{t
> }\mathbb{E}\_{P_t}\left[g\_\omega(X\_{t})\right] - \mathbb{E}\_{P\_{t+1}}\left[g\_\omega(X\_{t+1})\right] $ and $\lambda$ is a tunable parameter.
>
> This is then further relaxed to the following equation to drop the requirement on simultaneous access to all domains,
> \begin{align}
>     \min_\theta \max_\omega \mathbb{E}\left[\ell_{h_{\theta,t}}\left( g_\omega(X_{t}), Y_t\right) \mid Z_{1}^{t}\right] + \lambda L_{D_t} ,
> \end{align}
> where $L\_{D_t} = \mathbb{E}\_{P\_t}\left[g\_\omega(X\_{t})\right] - \mathbb{E}\_{P\_{t+1}}\left[g\_\omega(X\_{t+1})\right] $.
>
> We can see that the regularization term $L_{D_t}$ resembles the dual formulation of the Wasserstein-1 distance between two domains, thus we were referring to evaluating this regularization term as ``approximating the Wasserstein distance''.

---

> ### Comment · Reviewer_ybbK · 2022-10-14
> **Thanks for the response**
>
> I would like to appreciate the authors to carefully respond my comments. In particular, the authors' comments on monotonicity of $p$-th Wasserstein distance is pretty helpful, from which I learned some new concepts and ideas. In addition, the authors nicely incorporated more details of comparisons with related work. I think the quality of the manuscript has been improved a lot during the revision.

---

### Review · Reviewer_XcAN · 2022-09-17

**Summary Of Contributions:**

This paper focuses on supervised gradual domain adaptation, where the goal is to learn a classifier that performs well on a target domain given labeled data from a sequence of temporally related domains. This paper introduces generalization bounds for the supervised gradual domain adaptation setting. Analysis of these bounds points to the importance of minimizing the distance between sequential class-conditional distributions and the effect of the number of domains. Based on the bounds, a min-max learning objective is proposed and its effectiveness evaluated on the rotating MNIST, portraits, and FMOW datasets.

**Broader Impact Concerns:**

No concerns.

**Requested Changes:**

Requested changes:
- It would be useful to compare the results of Theorem 4.1 with the bounds of Kumar et al. (2020) for unsupervised gradual domain adaptation and Wen et al. (2020a) for multiple domain adaptation in order to understand the relationships among these settings. Presently these connections are only discussed at a high-level (e.g. exponential vs. linear relationship) but it is difficult to piece this together without seeing the other bounds expressed in similar mathematical terms.
- Clarify why the $W_\infty$ distance is too restrictive.
- Clarify the nature of assumption 3.3. The writing currently implies that all bounded functions are Lipschitz continuous ("we only assume that the empirical loss function is bounded and is hence Lipschitz continuous"). Also, the experiments use cross-entropy loss, which if I'm not mistaken, is neither Lipschitz continuous nor bounded.
- Explain in greater detail the various baselines (e.g. no adaptation, direct adaptation, etc.) and how the training domains are constructed.

Minor comments:
- Lemma B.1 has a typo in the $\sqrt{\cdot}$ expression
- In Theorem 4.1, it is not clear how the hypothesis $h_T$ is chosen.
- The definition of $L_D$ under (3) is difficult to parse. Is the minimization performed only over $t$? If so, then the $t+1$ should be removed from the minimization.

**Strengths And Weaknesses:**

Strengths:
- The supervised gradual domain adaptation setting is realistic and interesting.
- The generalization bounds provide insight into the relevant factors for learning in this setting.
- A specific objective is proposed based on the bounds and its performance evaluated on realistic datasets.

Weaknesses:
- The relationship to previous generalization bounds is not clearly demonstrated.
- Clarity of writing and explanation of details could be improved.

---

> ### Author Response · Authors · 2022-10-08
> **Response to reviewer XcAN**
>
> We thank the reviewer for the insightful comments and constructive suggestions. We shall address and clarify the major concerns raised in the following responses.
>
> ### 1. Detailed comparison to (Kumar et al. 2020) and (Wen et al. 2020)
> We have revised the related work section for a more detailed comparison, and we also included the final bound of existing works to illustrate the main differences.
>
> When compared to previous results on GDA, our result is only linear with respect to the horizon length $T$ and the average loss $h_{0}=\operatorname{argmin}_{h \in \mathcal{H}} \frac{1}{T}\sum\_{t=1}^{T} \ell\left(h(X\_t), Y\_{t}\right)$
> In contrast, the previous upper bound given by Kumar et al, which is for unsupervised gradual domain adaptation, is $O(\alpha_0\cdot\exp(T))$, with $\alpha_0$ being the initial error on the initial domain. In terms of the dependency on the time horizon $T$, ours is an exponential improvement.
>
> When compared to multiple-source domain adaptation, our result has the same order of error with respect to the Rademacher complexity (or the VC dimension) in supervised multiple source domain adaptation (MDA) (Wen et al. 2020), which is $O\left( \sum_{t}\alpha_t( \ell_{h_T}(X_t, Y_t) + \mathfrak{R}(\mathcal{F}) \right)$, where $\sum_t \alpha_t = 1, \alpha_t > 0, \forall t$ and $\mathfrak{R}(\mathcal{F})$ being the Rademacher complexity. Taking the weights $\alpha_t = \frac{1}{T}$, we can see that the error of a classifier $h$ on the target domain similarly relies on the average error of $h$ on training domains. However, note that our results scale with the averaged error of the best classifier on the training domains.
>
> ### 2. Discussion on the $W_\infty$ distance
> Previous results leverage the Wasserstein-infinity $W_\infty$ distance (Kumar et al. 2020) to capture the non-stationarity of gradual domain adaptation. In practice, however, this is rarely used and $W_1$ is more commonly employed due to its low computational cost. Moreover, even if pairs of data distributions are close to each other in terms of $W_1$ distance, the $W_\infty$ distance can still be extremely large with the presences of a few outlier data points from each domain. Hence, the use of $W_\infty$ in the context of gradual domain adaptation poses a (very) strong assumption on the changing distributions. Previous literature hence offers limited insights whereas our results work for the more general scenario.
>
> ### 3. Bounded and Lipschitz loss function
> We thank the reviewer for pointing out the mistake. In practice, our data are normalized and hence they are from a compact domain. Since the cross-entropy loss is continuously differentiable, its derivative norm is continuous and thus bounded on this compact domain. This hence implies the cross-entropy loss is empirically Lipschitz in our experiments.
>
> ### 4. More details on dataset constructions and baseline algorithms
> We have added more details to Appendix section C on the construction of the datasets and the baseline algorithms used.
>
> ### 5. Clarifications on typos
> We thank the reviewer for pointing out the typos. The typo in Lemma B.1 has already been fixed and the notations for $L_D$ have also been fixed. We have also clarified in Theorem 4.1 how the final classifier $h_T$ is obtained (we required the final classifier to be obtained by empirically minimizing the loss of each domain throughout the training horizon).

---

> ### Author Response · Authors · 2022-10-18
> **Thank you and we welcome further questions and comments.**
>
> We thank the reviewer again for the constructive feedback. We hope that most of the concerns have been addressed by this discussion. If there are any further questions and comments, we are very happy to follow up and discuss them.

---

> ### Comment · Reviewer_XcAN · 2022-10-27
> **Thank you**
>
> I would like to thank the authors for responding to each of my concerns.

---

### Review · Reviewer_DnNf · 2022-09-25

**Summary Of Contributions:**

Review for TMLR 401

Summary：
This paper studied the problem of supervised gradual domain adaptation, which arises naturally in applications with temporal nature. In this setting, the authors provide the first learning bound of the problem and the results are general to a range of loss functions and are algorithm agnostic. Based on the theoretical insight offered by the proposed theorem, this paper designed a primal-dual learning objective to learn an effective representation across domains while learning a classifier. The authors analyze the implications of the results through experiments on a wide range of datasets. In general, this paper makes certain contributions to the field of domain adaptation.



**Broader Impact Concerns:**

No concerns here.

**Requested Changes:**

Cons:

1. There might be missing literature regarding meta-learning. When we address the GDA, certain assumptions are required to ensure that GDA can be addressed well. The meta-learning based domain generalization methods might be good baselines. These methods can be changed to address the GDA problem as well. For example, you can train a good algorithm with MAML, then you can train models for target domains by using the data available in the target domain.

2. Like the first point, the meta-learning-based baselines are required to make the experiments more solid. Currently, the experiments are not solid. More baselines are needed.

3. It’s better to write down the problem setting using a definition, where necessary assumptions should be introduced.

4. CIFAR-10 or CIFAR-100 dataset should be added into the experimental section, for the benchmarking purpose.



**Strengths And Weaknesses:**

Main Review:

Pros:

1. This paper is solid and provides insightful theoretical analysis to the field. The proposed setting is practical and can be applied into many real-world scenarios, since the labelled data are available even in the target domains in the real world. Then, the related upper bound is proposed and motivates the proposal of the method, which is meaningful and contains certain contributions to the field of domain adaptation.

2. The whole paper is easy to follow in the view of techniques. The used math notations are clear. The reviewers enjoy reading the math part, which is clear and well descried.

3. Notations are consistent, no much confusing points from theorems or proofs themselves.

Cons:

1. There might be missing literature regarding meta-learning. When we address the GDA, certain assumptions are required to ensure that GDA can be addressed well. The meta-learning based domain generalization methods might be good baselines. These methods can be changed to address the GDA problem as well. For example, you can train a good algorithm with MAML, then you can train models for target domains by using the data available in the target domain.

2. Like the first point, the meta-learning-based baselines are required to make the experiments more solid. Currently, the experiments are not solid. More baselines are needed.

3. It’s better to write down the problem setting using a definition, where necessary assumptions should be introduced.

4. CIFAR-10 or CIFAR-100 dataset should be added into the experimental section, for the benchmarking purpose.

---

> ### Author Response · Authors · 2022-10-08
> **Response to reviewer DnNf**
>
> We thank the reviewer for the insightful comments and constructive suggestions. We address and clarify the major concerns raised in the following sections.
>
> ### 1. Comparison with Meta-Learning
> We agree that it is important to provide an extensive evaluation of our algorithm with various baselines, thus we have included a comparison of our method to those based on meta-learning with evolving data distribution such as EAML [1] in Table (2). However, we want to note that such a comparison may not be fair, since (i) our GDA method only has access to two data distributions at one time while meta-learning methods have access to all tasks (all data distributions) and (ii) GDA's objective is to adapt to only one target distribution, while meta-learning methods need to be prepared to adapt to more objectives (e.g. EAML is designed for adapting to a changing target distribution). We have also added a section to further analyze our results and compare the details of the two methods.
>
> [1] Liu, Hong and Long, Mingsheng and Wang, Jianmin and Wang, Yu, Learning to adapt to evolving domains, Advances in Neural Information Processing Systems, 2020.
>
> ### 2. Comparison with CIFAR10 and CIFAR100
> We agree with the importance of benchmarking our algorithm, though we want to kindly point out that CIFAR10 and CIFAR100 have not been used as standard benchmark datasets for domain adaptation. While rotating MNIST has been widely used for assessing the adaptation ability of different adaptation algorithms (e.g. it is also used in [1]), the CIFAR datasets are rarely used for this purpose. In fact, it is unclear what is the best way to modify the CIFAR dataset for the (gradual) domain adaptation task. For example, the images of planes in the CIFAR datasets may already contain planes of different orientations. In this case, rotating the image may not be reasonable for creating a gradual domain adaptation task, as it may not be able to reflect the nature of gradually evolving distributions.
>
> ### 3. Stating the problem setting as a definition
> We agree that this would greatly help to clarify the setting and various assumptions, thus we have restated the setting as Definition 3.2.

---

> > ### Comment · Reviewer_DnNf · 2022-10-08
> > **All concerns are addressed**
> >
> > Thanks for addressing my concerns. I will recommend accepting this paper now.

---

### Decision · Action_Editors · 2022-11-02

**Recommendation:** Accept as is

**Comment:**

This paper studies the supervised gradual domain adaptation problem. Theory and algorithms are provided. More specifically, the authors have proven that the supervised gradual domain adaptation has a smaller upper generalization bound than that of unsupervised gradual domain adaptation. Effective algorithms are inspired by the theory. All reviews appreciate the contribution and we therefore recommend an acceptance.

**Audience:**

Domain adaption is a hot topic and has many audiences both from academia and industry.

**Claims And Evidence:**

The claims are supported by theory and empirical results.

---

> ### Author Response · Authors · 2022-11-14
> **Thank you to editors and reviewers**
>
> We want to thank the reviewers and the editors for their constructive comments during the review process. Based on the comments received, we have uploaded the camera ready version.